# Living Conditions of Adolescents Who Have Attempted Suicide in Mexico

**DOI:** 10.3390/ijerph17165990

**Published:** 2020-08-18

**Authors:** Rosario Valdez-Santiago, Alma Lilia Cruz-Bañares, Anabel Rojas-Carmona, Luz Arenas-Monreal

**Affiliations:** 1National Institute of Public Health, 62100 Cuernavaca, Morelos, Mexico; rosario.valdez@insp.mx (R.V.-S.); luz.arenas@insp.mx (L.A.-M.); 2Independent researcher, 54060 Estado de México, Mexico; almaliliacruz@gmail.com; 3Postgraduate Program in Public Health, Faculty of Medicine, Federal University of Ceará, 60430-160 Ceará, Brazil

**Keywords:** social determinants of health, attempted suicide, adolescence, Mexico

## Abstract

Suicidal behavior represents a complex public health problem, with a rising number of suicide attempts registered among Mexican adolescents. We undertook a qualitative study in order to understand the living conditions of adolescents who had attempted to take their lives in five Mexican states. We interviewed 37 adolescents who had engaged in suicide attempts in the year prior to our study. To code and analyze the information, we defined the following three categories of living conditions as social determinants of health for adolescents: poverty and vulnerability, education, and health care. To this end, we followed the methodology proposed by Taylor and Bogdan, and used Atlas.ti 7.5.18 software for analyses. Among our findings, we noted that poverty, manifested primarily as material deprivation, rendered the daily lives of our interviewees precarious, compromising even their basic needs. All the young people analyzed had either received medical, psychological, and/or psychiatric care as outpatients or had been hospitalized. School played a positive role in referring adolescents with suicidal behavior to health services; however, it also represented a high-risk environment. Our findings highlight the urgent need to implement a national intersectoral strategy as part of comprehensive public policy aimed at improving the health of adolescents in Mexico.

## 1. Introduction

The complex phenomenon of suicidal behavior poses a critical public health problem. With approximately 800 thousand individuals taking their lives every year, suicide has been identified as the second leading cause of death worldwide among young people between 15 and 19 years of age [1].

Studies in Mexico have reported a systematic increase in suicidal behavior—particularly suicide attempts—among adolescents [2,3,4]. According to Borges et al., suicide deaths spiraled by 275% from 1970 to 2007, particularly among those in the 15–29 age bracket [4].

Given their complexity and their relevance as a precursor of death by suicide, we sought to gain a better understanding of suicide attempts from the standpoint of the social determinants of health (SDH). These have been defined as the “circumstances in which people are born, grow, live, work and age, and the wider set of forces and systems shaping the conditions of daily life… including the health systems. These circumstances are shaped by the distribution of money, power and resources at the global, national and local levels,” as well as by the policies implemented in this regard [5].

According to Viner et al., the principal SDH for the adolescent population include national wealth, income inequality, access to education, and support from the family, school, and peers [6]. These determinants have been associated with suicide attempts. Various authors have indicated that precarious work conditions, occupational stress and low income among parents are factors contributing to suicide attempts in adolescents [7,8,9]. As social determinants, health systems influence suicidal behavior when lack of access, limited availability and low quality of health services are barriers for the prevention or early detection of this public health problem [1].

The Pan American Health Organization (PAHO) and the Economic Commission for Latin America and the Caribbean (ECLAC) have reported that the Latin American and Caribbean region suffers from one of the highest levels of inequality in the world, directly affecting health conditions [10,11].

Beginning in the 1980s, Mexico adopted neoliberal policies focused on reducing social spending, leading to an increase in the rates of poverty and social inequality [12]. In 2018, 41.9% of the population lived in poverty, and 7.4% in extreme poverty [13]. In that same year, a new administration took office in Mexico. Unfortunately, improving social conditions in such a short time is not possible [14].

The SDH perspective allows for clearly understanding how social inequality impacts the living conditions of adolescents in the five Mexican states we analyzed. Among its most notable characteristics are an unequal distribution of material resources and the differentiated access to health services and schooling. From the SDH standpoint it is also possible to perceive how these conditions have set the stage for the high prevalence of suicide attempts among Mexican adolescents.

In light of the above, we undertook the present study to understand the living conditions of adolescents who had attempted to end their lives in five Mexican states.

## 2. Materials and Methods

This work is part of a broader study entitled Suicidal Behavior among Young Adolescents in Mexico, a mixed-methodology initiative approved by the Research Ethics Committee of the National Institute of Public Health (CIE 1471). Its qualitative component includes analyses of (a) the health system response to suicidal behavior in adolescents and (b) the life experiences of adolescents who have attempted to take their lives. This article pertains to the latter.

The methodology we followed, based on the three dimensions proposed by the Consolidated Criteria for Reporting Qualitative Research (COREQ), is described below: [15] (a) research team and reflexivity; (b) study design and (c) analysis and findings.

### 2.1. Research Team and Reflexivity

The purpose of understanding suicidal behavior through a qualitative approach arose from a prior quantitative research experience, the analysis of the suicidal attempt component, in the questionnaire for adolescents, of the National Health Survey (ENSANUT 2012), performed in Mexico. In this study, we documented the national prevalence of suicidal attempt and its related factors [16]. However, our interest grew when we reflected on the fact that this is a multidimensional phenomenon. We set out to further explore the living conditions of adolescents who attempted suicide, and in this way, to broaden our understanding of the complexity of the problem, as an illness-health-attention process, within a health social determinant framework. Furthermore, to analyze it from another perspective that would allow us to go beyond statistics, the clinical conditions of the persons involved, or individual factors (e.g., age, substance use, gender), traditionally reported in international literature.

The authors form part of a broader multidisciplinary research group. The team responsible for the interviews was composed of one physician and six psychologists—three with clinical backgrounds. All were experienced in conducting semi-structured interviews. Four team members held doctorate degrees in health-related fields including medical anthropology, psychology, and public health, and three held Masters in public health.

The team of interviewers included six women and one man, and ages ranged from 35 to 60 years. All had experience in public health and were specialized in health promotion, mental health and vulnerable populations: children, youth, the indigenous population and women. The first contact between the research team and participating adolescents occurred on the day of the interview.

The team maintained an ethical position vis a vis interviewees and local mental health authorities throughout the research process. 

### 2.2. Study Design

The nature of the subject of our study led us to select a qualitative approach, with hermeneutic phenomenological design [17], with the objective of understanding thought interpretation the essence of the lived experiences of a group of people surrounding a phenomenon, in this case, living conditions experienced by adolescents who attempted suicide.

An interview guide was prepared, as an instrument for obtaining information, based on an expert revision. Two interviews were performed as a pilot, and adjustments were made. Likewise, the interviewers’ team held meetings, in order to have a common perspective when collecting information. The purpose of the interview was to “know the personal and contextual aspects reported by the adolescent that lead him/her to a suicidal attempt, in addition to the use of health services”. The selection of the participants happened at two points in time: (1) selection of the participants’ states and (2) selection of persons to be interviewed.

(1)Participating states were selected through convenience sampling according to the following criteria: (a) they maintained a mental health program, including care for suicidal behavior, and (b) health authorities were willing to commit to the identification and monitoring of participating adolescents. On this basis, we selected five states, in three different regions of Mexico: (a) Baja California Sur, (b) Aguascalientes, (c) Morelos, (d) Tabasco and (e) Campeche (Figure 1).(2)Regarding the selection of persons to be interviewed, the inclusion criteria were: (a) To be an adolescent, understanding that adolescence is the period of human growth and development that happens after childhood, and before adulthood, and which corresponds to the ages between 10 and 19 years old [18]. (b) Have attempted suicide in the year prior to the interview (2017–2018); (c) Receive some type of mental health care from the public services in his/her locality, and (d) Agree to participate in the study.

For the selection of the participants, we worked in collaboration with the mental health personnel in each state, who identified in their clinical files persons who met the inclusion criteria defined by the research team. They informed the purpose of the study, by telephone, to the adolescents’ relatives, they explained the invitation to be interviewed and the importance of their participation. If they accepted, the interviews were scheduled according to the most convenient time for the person to be interviewed and the accompanying relative.

Based on this information, two interviewers from the team traveled to each state to perform the field work, following the appointment schedule, previously agreed with the mental health personnel of the state. In general, the interviews were performed at the mental health clinics. The interview was performed in private, face to face. In all cases, the written consent from the adolescent was requested and the accompanying adult signed an informed consent, prior to the interview, in order to perform and record it. The approximate duration of each interview was 60 min, only once. During the interview, both interviewers took notes in a free-format field diary.

The couple of persons responsible for the interview always included a psychologist with clinical experience to identify and resolve possible emotional reactions during the interview.

We decided to interview all the persons that the local team could recruit; therefore, a theoretical saturation criterion was not applied, in this way, adolescents with suicidal attempt from five states in the country were interviewed. In some cases, the adolescents missed their appointment, and the contact was lost, for unknown reasons. Only in one of the participating states, mental health personnel guaranteed transportation from the adolescent’s home to the clinic in an official vehicle from the health services. Other adolescents were interviewed, but were not included in this study, because they did not meet the suicidal attempt criterion.

In all participating states, it was agreed with the mental health personnel to ensure post-interview follow-up of all the persons interviewed, in order to guarantee their safety and integrity.

### 2.3. Analysis and Findings

All the interviews were transcribed for analysis purposes, assuming confidentiality ethical principles.

Having as explanatory framework the SDH and with the purpose of analyzing the living conditions of the adolescents who attempted suicide, we used the method proposed by Taylor and Bogdan [19] for the analysis. This method encompasses the following three steps: (1) Discover; from multiple readings of the transcribed interviews individual notes were taken as well as relevant quotations were selected, we identified 7 emerging themes: 1. Population profile, 2. Family economy, 3. Housing, 4. Schooling, 5. Occupation, 6. Free time, and 7. Health care. We started organizing how we wanted our findings to be presented. (2) Coding; we developed the emerging themes as coding categories and described each one in a coding manual. Each author coded different categories and they were peer reviewed. Atlas.ti 7.5.18 software was used for this purpose, and (3) Discounting; from coded information and after team discussions, re-readings and literature reviews, we defined the interpretation on the basis of 3 specific categories that gave account of the social determinants, (a) Poverty and vulnerability, (b) Being in school: protection or vulnerability, and (c) Accessibility to mental health services. These categories led our analysis and the structure of our article.

For reporting our results, we have included seven quotations, each one indicating the sex and age of the adolescent who shared the information. All the data reported are consistent with our findings and with the objective of this article. For each of the categories presented we included the main findings as well as minor themes that contribute to a better understanding of the situation and its relationship with the SDH.

## 3. Results

### 3.1. Characteristics of the Adolescents Interviewed

We interviewed 37 adolescents: 14 male, 22 female, and one male transgender, with an average age of 16.5 years old (age range 13–19 years old). Of these, 59.50% were students, 5.4% combined studies with paid work, 13.5% had paying jobs, but did not study, and 11% neither studied nor worked. As many as 2.7% were in drug rehabilitation programs, and 8.1% had an unknown occupation. As regards health services, 57% have health coverage, and 11% do not, while the status of 32% was unknown.

### 3.2. Poverty and Vulnerability

One dimension explored in our study of adolescents was poverty, manifested as material deprivation. We found that most breadwinners had precarious occupations with low income, long work hours, and rotating shifts, such as: drivers, police officers, laborers, street vendors, and small business owners. In a few cases, we found that the main household providers practiced professions such as teaching or nursing. In these homes, despite financial difficulties, family members had access to activities such as eating out and going to the movies, among other things.

The job insecurity of the persons responsible for the family, not only represents a low income, which usually does not cover all the basic needs of the family, but it also means little time for family life, inadequate performance of their mentoring roles towards the adolescents, it makes it difficult for the persons who provide it to rest, compromising their quality of life and that of their families, due to the resulting implications.

Another aspect reflecting the precarious nature of the daily lives of our interviewees was having to share material resources and food with members of the extended family. Although not common, we encountered families in which as many as ten people shared resources intended for food.

Adolescents perceived material scarcity within their families, and in some cases, this led them to develop coping strategies, and to seek some sort of paid work to buy personal items and help with family expenses. This situation caused tension and conflict. This is the case of an adolescent woman, whose motivation to contribute to the family caused discomfort, especially to her father, who felt threatened his role as provider.

This kind of insecurity affected adolescents in various ways: family conflicts, violence, inadequate food, and sleep, whether because of the lack of adult supervision of their hours, or because of the scarcity of consumer goods (food) for meeting their basic food requirements.

Furthermore, these situations affect their possibilities for socialization, recreation or rest, whether due to the location and characteristics of their place of residence, little availability of recreation areas, or since it is necessary for them to have a paying job, they do not have time for recreational activities. Such difficulties may facilitate adopting pastimes, such as excessive use of electronic media and, therefore, to isolate themselves, which can result in adverse effects to their physical and emotional health. 

Those in the north of Mexico (Baja California Sur), referred to frequently visiting the waterfront and enjoying outdoor activities such as skating, bicycling, or walking alone or with friends on the beach. Meanwhile, those who lived in states in the center of the country (Aguascalientes and Morelos), spoke of visiting the public park in the middle of the city to walk or get together with friends, as well as watching television or playing videogames. Finally, those living in the southeast (Tabasco and Campeche), referred to going to the movies, walking along the waterfront, and playing soccer or volleyball.

The following quotations illustrate the above:


*“It’s just that my mom would always say that she didn’t have any money. Then, since she was in the Sin Hambre (The Mexico Without Hunger National Program (2014–2018) was an Inclusion and Social Welfare Strategy of national scope, which aimed to ensure food security and nutrition for the 7.01 million Mexicans living in extreme poverty.. https://plataformacelac.org/politica/). program, well they’d give us a package of tuna or a bag or rice. There wasn’t a lot, and because we were many, if so and so ate a lot, then so and so wouldn’t eat at all.” (Female, 15 years old).*



*“… sometimes he [the father] yells at me: why am I working. Or sometimes he yells really horribly at my mom: why does she let me work. [He says] that’s what I have my father for, for him to maintain me, and so forth.” “…I don’t know what I did to make him angry; I just started working because I wanted to buy my own things, so as not to by a bother…”. (Female, 15 years old) [The underlining is the authors.’]*


### 3.3. Being in School: Protection or Vulnerability

School represented an environment that could help protect adolescents, or one that increased their vulnerability. School represented an environment that could serve to protect adolescents or one that increased their vulnerability. School was significant in the lives of adolescents, facilitating their academic development, their relationships with peers, and even their future plans.

School was the main setting for meeting and learning to get along with peers. Relationships forged there could help adolescents cope with stressful situations, but could also lead to violence, whether as perpetrators or as victims. Teasing, criticism, physical stereotypes, couple relationships, and even punches or sexual harassment, were reported by some adolescents. These types of situations had consequences at the individual, family, and school levels, in many cases being the trigger for self-harm or suicide attempts.

Relationships with teachers sometimes also served as a protective factor, especially since teachers were often those who referred students to health institutions. In these cases, many adolescents acknowledged the help they received and its impact on their personal and scholastic development. On the other hand, several adolescents complained that teachers and staff lacked protocols for stopping conflict, despite of being aware of the bullying and violence that some students were experiencing.

Adolescent academic performance varied widely. In the face of stressful situations encountered within the family and school contexts, some students were able to pass their courses without difficulty, while others found their grades negatively affected. Some of our interviewees reported that they had failed, been forced to repeat a school year, or had decided to leave school for a while. The interest of parents in what occurred at school was not always constant. We found that in some instances, adolescents identified school as the setting in which they began to harm themselves.

While not all participants were attending school at the time of the interview, the majority expressed interest in continuing their studies. They perceived schooling as key to achieving other life goals.

The following quotations illustrate the above:


*“You can say that I suffered from bullying back then. In those days, apart from them hitting me, the others laughed at me. They didn’t care if the other guy was hitting me. They laughed because I couldn’t take it.” (Male, 16 years old).*



*“In middle school, in second year, the director [female], the first week I was there, kind of noticed that, like, something wasn’t right in my life, and told my mother that there was a, I don’t know what the name of it is, the DIF (The DIF is the System for Comprehensive Family Development. This is a federal agency that provides various services for the vulnerable population, among them psychological counseling https://www.gob.mx/difnacional) or something like that, nearby, and that they had a psychologist [female] there.” (Male, 17 years old).*



*“It was in the bathroom with a friend. I was watching her cut herself and I asked her why she did that, what the reasons were. And then she says, ‘It’s a tranquilizer. It relaxes you.’ Next, she said, ’Do you wanna try it?’ Then, well, curiosity gets the better of me, and I tried it, and I started to like it. I didn’t do it every day, just when I had problems at home.” (Female, 15 years old) [The underlining is the authors’]*


### 3.4. Accessibility to Mental Health Services

In Mexico, all adolescents have the right to health-care services through the public sector. They are protected in one of three ways: as direct beneficiaries by reason of their enrollment in a public school, through their family employment, or through the Seguro Popular (The social protection system known as *Seguro Popular*, the predominant health insurance strategy of the Mexican government from 2000 to 2019, included treatment for the principal mental disorders. The current administration eliminated it and, in its place, created the Health Institute for Well-being (*INSABI* by its Spanish initials), in charge of providing services to the population not covered by private or public Social Security institutions.) insurance scheme, intended for those lacking Social Security coverage.

All adolescents interviewed had received health-care services from public health facilities as the result of their suicide attempt, either as outpatients (medical/psychological/psychiatric) or in hospitals (medical/psychiatric); some had used the emergency services.

All adolescents had received psychological or psychiatric services at the primary-care level. Educational institutions were of enormous support in assisting adolescents at risk. Some participated in mental health programs coordinated by local health services, and the teaching staff often identified those adolescents needing referrals to health services for psychological counseling.

In some cases, adolescents mentioned that thanks to the medical attention they received, they felt better, they identified areas of opportunity in their lives and learned different ways of facing the stressful situations they encountered. In these cases, they mentioned that they had established good communication with their therapist and expressed their gratitude for the service received.

Another aspect that we identified concerned the barriers encountered by participating adolescents in obtaining care from these services: (1) appointments scheduled less frequently than what they consider they need; (2) health facilities were geographically removed from their homes, and transportation imposed time and financial pressures; (3) appointment schedules overlapped with school schedules; (4) dissatisfaction with the psychological or psychiatric attention received; and (5) frequent changes in the psychology and psychiatry personnel that provide the service.

These difficulties may mean that adolescents perceive that they do not receive an appropriate attention and follow up of their mental health, which, according to what they refer, may lead them to question importance to continue attending.

Even if attention was outpatient care in most of the cases, some adolescents were hospitalized after a suicide attempt. In the case of psychiatric institutions, we identified that their perception about the attention they received is related to various aspects that they consider that difficult their recovery: (1) the design of hospital rooms without exterior windows and with artificial light; in some cases these rooms were air conditioned (because of the tropical climate in these regions), to such extent that the rooms were cold; (2) the institutionalized routine; and (3) having to share rooms with seriously ill patients, a situation hardly conducive to their recovery. On balance, the care received in psychiatric hospitals served to stabilize the situation that had given rise to the suicide attempt.

For those adolescents that used the emergency services, the experience was emotionally significant. The family was, in some cases, an indispensable element to access timely medical attention. Whether the adolescent was taken directly to the hospital, or if an ambulance was requested through the emergency number (911), the adolescents reported their experience once they were admitted to the emergency service. Self-reports on emergency hospital treatment referred to unpleasant medical procedures, such as having their stomachs pumped, or the simple fact of being hospitalized. It is worth noting that none of the adolescents indicated inadequate care by hospital personnel; on the contrary, they described displays of sensitivity. However, in most of these cases, the care received was directly related to the medical emergency at hand. Furthermore, no psychological/psychiatric counseling nor follow-up support was offered to them after leaving the emergency facilities.

Both in outpatient and hospitalization services, most of the families of the adolescents did not pay for these services, even though in some institutions such services were not free. However, all the adolescents and their families incurred transportation and other expenses to use the health services. Aside from those treated at Social Security facilities, families were required to purchase the medicines they needed.

Health systems, as social determinants of health, are a crucial element for access to services and their quality, in this case, the mental health services. Outpatient services may be a window of opportunity to identify and appropriately treat mental health conditions in adolescents, or on the contrary, they can become a barrier that contributes to suicidal behavior, or suicide, in this population group.

The following quotations illustrate the above:


*Interviewer “…Did you see another psychologist?*



*Adolescent: Yes, in fact I went to see a psychologist only once, because she started to tell me that I was exaggerating and that I had to realize, “look at me” she said, that she was making a living, very arrogant. She told me that I was exaggerating and that I should realize that I was harming my parents, that this or that was my fault” …. (Female, 17 years old).*



*Interviewer: “How long were you there?”*



*Adolescent: “Ten days. It was the worst thing in the world that can happen...It’s horrible, not because the treatment’s bad; it’s horrible. If you’re used to going out and things like that, there, they have you locked up. And instead of windows it’s skylights. There’s artificial climate all the time, and it seems that they have it at 18ºC because it’s cold all the time. They make you take a shower real early, at about 6-7 in the morning. And if you don’t take a shower, [then] you’re out of luck. You must behave well because if you don’t, they tie you up or they give you a shot with something so that you go to sleep and calm down and things like that. I never got to that point, where they had to tie me up or things like that. But, did you see that? Yes, and there are people who, well… you’re there and you realize that those people are worse off than you are. And I don’t know. It’s bad.” (Female, 15 years old).*


## 4. Discussion

This article contributes to an understanding of the relationship between life conditions and suicide attempts in Mexican adolescents.

Our study demonstrated that the families of the adolescents interviewed were caught in a cycle of employment insecurity and poverty, expressed in daily life. Viner notes that low levels of national wealth, income inequality and limited access to education are among the determinants affecting the health of the adolescent population [6].

Furthermore, poverty is closely linked to insecurity in the areas of employment, social conditions, and income [20], negatively affecting access to health services. More specifically, some studies have shown that employment insecurity among parents contributes to suicidal ideation among the adolescent population [7,8,9]. A study in Mexico indicated that adolescents considered family problems the main cause of attempted suicide [21].

Even if having access to education is a protective determinant during adolescence, experiences lived within the school context play a double role, offering both risk and protection. Echoing findings reported by Sharma [22], our study found that harassment and school fights were common experiences for adolescents who attempted suicide. The design and execution of protocols for preventing and addressing harassment at school, involving all actors in school institutions, including family, and which are coordinated with health services must be a priority [23]. These protocols must be aimed at those who watch the harassment, those who live it, and those who exercise it and live it, being the latter those who experience the highest rates of negative mental health outcomes, including depression, anxiety, and thinking about suicide.

Conversely, school provided protection when teachers and schoolmates were allies [24]. Thanks to the participants that experienced the school protective role in prevention or timely attention, the need to have a sense of connection between the adolescents and responsible adults, who belong to the school, is reinforced. This connectivity process provides support to those adolescents that are facing social and emotional problems [23]. The need to create spaces that provide tools to the students, not only in the academic aspect, but also emotionally and socially, arises as part of this process [25].

We noted the lack of education programs aimed at preventing and developing life skills. However, the literature shows that there are successful experiences around this type of strategies, such as the so-called Gatekeeper training in school environments, which incorporates different programs that seek to develop in participants (students, teachers, families), the knowledge, attitudes and skills to identify those persons at suicide risk, the severity, and to refer them to specialized services, if necessary [26,27]. For his part, in Mexico, Chávez A., showed positive results in the application of a psychoeducational program for preventing suicide in young people [28].

About the access to and use of health services, we found areas of opportunity, both to identify adolescents at risk and, once identified, for them to receive the required attention, in order for the risk of future attempts to decrease. From participants’ feedback and comments, we identified the importance of strengthening the training of health professionals (medicine, psychology, nursing) in two ways: first, by reinforcing their knowledge related to this problem, within the prevention and risk identification areas [29] and second, to achieve an ongoing relation of trust with the adolescents that attend the services (mainly psychology) [30] There is evidence that states that adherence to and conclusion of psychological therapy may help in preventing a completed suicide, or further attempts [30].

Likewise, considering the low use of health services by adolescents, who have been reported to have attempted suicide by Borges [31], the need to strengthen the processes to identify adolescents at risk (due to risk factors or previous attempts) was revealed, this must be done by developing synergy not only with schools, but also with other meeting places, and using programs such as national or local emergency telephone lines (911). All these strategies require having an attention and follow up protocol that guarantees that the adolescent receives all the required attention, otherwise, the effectiveness of the strategy cannot be proven [32].

Our findings regarding life conditions, revealed the importance and lack of free time as a source of well-being and part of the right to recreation [33]. Due to economic shortcomings, lifestyle, unmet basic needs, lack of space, time for extracurricular activities in education institutions, and absence of infrastructure, the adolescent population interviewed lacks physical space and time for activities that promote their health and development.

A study conducted in Mexico found that “passive activities—watching television, using the computer and navigating the social networks—did not stimulate creativity, but rather almost doubled the risk of suicide among adolescents” [34]. Among our interviewees, these were the most common activities.

In our study, as well as in the work of Oliveira and Rosa, we can see that free time and recreation are of secondary importance compared to studies or work [35]. Nevertheless, in line with the PAHO recommendations [1], we reaffirm the importance of free time as an element of future interventions for the prevention of suicidal behavior.

## 5. Conclusions

Public policies reflecting the complexity and comprehensive nature of the health needs of adolescents are urgently required; these must effectively pursue the material and social empowerment of their families. As indicated by the Organization for Economic Co-operation and Development (OECD), simple policies will not make a significant difference; instead, a combination of cross-cutting policies for simultaneously providing protection and reducing risk is needed [36].

At a national level, economic growth is crucial for improving the quality of life of people, as well as to ensure education and health for the population. The Mexican National Development Plan (2019-2024) [37], includes a strong component of social policy matters, for various underprivileged populations in Mexico, with the purpose of generating welfare. Regarding the health system and access to mental health attention, we consider it necessary and urgent to strengthen the current health reform, in such way that it influences the reorganization of public health services, and an increase to the allocated budget [38]. Specifically, in order to generate actions aimed at preventing suicide, it is necessary to create a Mental Health Law in Mexico.

In this regard, political will represents an important component for possible changes that can impact the SDH and life conditions of the population in general, and the adolescent population specifically, however, this is not enough.

In the school context, it is crucial to strengthen partnerships [39] for the prevention and management of suicidal behavior. To this end, the educational sector [40] must participate in the implementation of a national, intersectoral strategy within a comprehensive public policy for the adolescent population.

We can confirm that when we approach the suicide attempt qualitatively, analyzing life conditions and social determinants in depth, the edges, and the interaction of all the dimensions involved is evident. It is necessary to continue visualizing and going deeper in the understanding of this public health problem from this approach. We must design multisectoral strategies as part of a comprehensive approach allowing adolescents to express their sense of “being-in-the-world” [41], always taking into consideration that this phenomenon is influenced by different aspects; therefore, there is no single way to face it. The integration of different strategies adapted to the needs will help to face it successfully [42].

## Figures and Tables

**Figure 1 ijerph-17-05990-f001:**
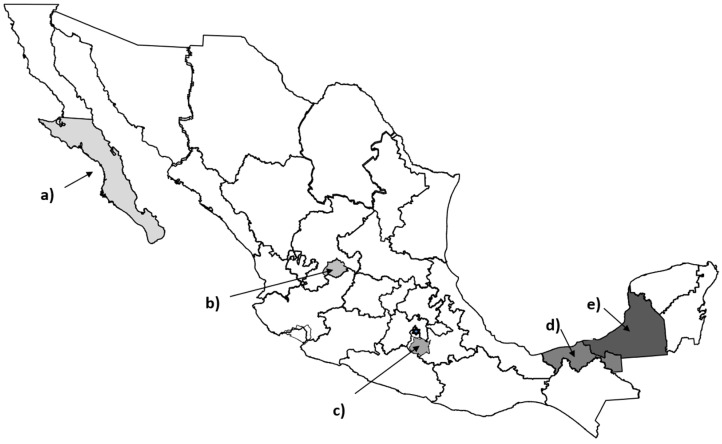
Participating states: (**a**) Baja California Sur, (**b**) Aguascalientes, (**c**) Morelos (**d**) Tabasco and (**e**) Campeche.

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
