# Peer review of "Living Conditions of Adolescents Who Have Attempted Suicide in Mexico"

_ijerph, 2020, doi:10.3390/ijerph17165990_

Round 1

Reviewer 1 Report

Thank you for the opportunity to review this article. As it is stated it contributes to an understanding of the relationship between life conditions and suicide attempts among Mexican adolescents.

The qualitative nature of the article is of special importance, as these are frequently missing in the literature. 

Nonetheless, the text would in my opinion benefit from an updated literature review, as for example WHO (2014) report states 800.000 deaths per year due to suicide and not one million as cited in the study.

The most important part where I feel the paper needs improvement are the method and results. Qualitative research is not simply describing a few themes, instead it means compiling themes after detailed coding into a specific way of understanding the person/circumstances behind it. Usually there are some models that represent these understandings. The article stays on a too superficial level, or at least does not do the justice to the qualitative data collected and to the topic. I feel article needs more in depth descriptions of the processes behind the interviews and its analysis. I would prefer for the analytical part to be more informative and transparent about, describing the ways analysis was done (from the reference of Taylor and Bogdan it is difficult to know what was performed, for example grounded theory, thematic analysis, IPA, ...). COREQ standards could be used for making the article more reliable. 

I would suggest to prepare the text with the suggested improvements.

Author Response

Dear reviewer,

We thank you for your comments and for the time you took to review our manuscript. All your observations were very enriching, and they certainly contributed greatly to the presentation, organization, contents, and reflections of the work performed.

Based on the suggestions and changes, we deleted the first part of the title, remaining as follows: Living conditions of adolescents who have attempted suicide in Mexico

In attached file, you will find the answers to your requests.

Reviewer 2 Report

This is such an important area of research and I like the way that you frame this within a social determinants approach. You set up an interesting argument about the relationship between poverty and school and youth suicidality. I recommend taking a stronger focus on poverty and school as two important social determinants and structure your whole paper around these two determinants (ie. set out policies and indicators in these areas in your introduction, emphasise these in results, and align your discussion and conclusion specifically to school and poverty).

Introduction is engaging and sets out a clear rationale for the research question and approach. However, some more detail about health, mental health and youth policy in Mexico would be useful to provide context.

Materials and methods section needs substantial work. The authors state that "..we analyze the interviews conducted with adolescents who attempted to take 72 their lives in the year prior to the study..". However, the methods take the data beyond analysis and into interpretation and recommendations.

You need to include detail about how young people were recruited and, in particular for this subject area, how young people's voluntary consent was sought and how risks associated with distress caused by the interview were managed. It would help to have more detail about the authors' experience and skills in this area - what made them qualified and safe interviewers for these young people. What were the inclusion and exclusion criteria for participants? How you define 'adolescents' (what age range)?

You need more detail about the questions asked in the interviews.

Results

3.3 Health care: outpatient services vs. psychiatric hospitalization: This section does not seem to align with your focus on social determinants and their influence on suicidality. It is about what happens after the suicide attempt.

Section on school was really interesting - education is a relevant social determinant.

You need to explain how free time relates to suicidality. The connection here is not clear.

Discussion: There are some important points raised here that reinforce the importance of a social determinants approach. However, the structure and presentation could be clearer. I suggest some clear thematic subheadings to set out the discussion.

Conclusions: There isn't a logical enough flow from discussion to conclusions. The conclusions are too general. You have illustrated important links between poverty and suicidality. What are your conclusions here? You have set up an effective argument about social determinants of health, but haven't sufficiently set out the conclusions in terms of a social determinants framework. The exception is the theme of school, which is good.

Author Response

(The authors gave the same response as above.)

Reviewer 3 Report

TITLE

  1. The quote used is a bit confusing – if you are focusing on attempted suicide – are you trying to highlight factors associated with resilience or are you stating the individual’s life took a turn for the worse when things got difficult?

The word “change” is neutral and so it is confusing.

ABSTRACT

Very good Abstract

INTRODUCTION

  1. Are there any prevalence statistics to report to demonstrate the rates have been on the rise?

METHODS

  1. What was the method for recruitment? Was it through mental health programs? I think more details need to be offered. Were they referred/contacted by their social worker or mental health care provider? Was incentive provided? Were posters put up in these cities or was social media used?
  2. Were only interview (qualitative) questions used to assess your variables? Or did you also employ scales and surveys to determine: (a) poverty and vulnerability, (b) health care, (c) education and (d) free time?
  3. What was your definition of adolescent – what age range did you use?

RESULTS

  1. Better to identify the participants by their gender rather than biological sex (participants identified as men, women, and transgendered)
  2. In the characteristics, report the average age and age range of participants
  3. The quotes support that the adolescents live in economic precariousness. In the manuscript text try to link that with mental health issues and factors that could lead to attempted suicide.
  4. “On balance, 176 the care received in psychiatric hospitals served to stabilize the situation that had given rise to 177 the suicide attempt, but the experience left adolescents with lasting emotional scars.”

“On balance…” is awkward

  1. The quotes are very good – but I feel there are too many of them. Is it possible to restructure the Results section so you have more of a summary from the authors and 2-3 quotes under each subheading to illustrate your points?
  2. None of the quotes captured specifically talk about suicide or suicidal ideation. If this paper is focused on kids who have attempted suicide – then more direct quotes on this theme need to be shared.

DISCUSSION

  1. “According to the Pan American Health Organization (PAHO) and the Economic 301 Commission for Latin America and the Caribbean (ECLAC), the Latin American and Caribbean 302 region suffers from one of the highest levels of inequality in the world, directly affecting health 303 conditions [12, 13]. 304 Beginning in the 1980s, Mexico adopted neoliberal policies focused on reducing social 305 spending, leading to an increase in the rates of poverty and social inequality [14]. In 2018, 41.9% 306 of the population lived in poverty, and 7.4% in extreme poverty [15]. In that same year, a new 307 administration took office in Mexico. Unfortunately, however, this has not yet been reflected in 308 an improvement in social conditions [10]. 309 Poverty is closely linked to insecurity in the areas of employment, social conditions and 310 income [16], negatively affecting access to health services.”

This is better suited to the Introduction

  1. You can add that more funding needs to be poured into creating safe spaces and recreation opportunities for young people
  2. Can recommendations be added for the education sector – for example, augmenting mental health services offered within school or increase mental health training of educators or anti-bullying programs for students
  3. Are there any successful programs that have been implemented in Mexico? Or internationally that can be implemented in Mexico? Are there SDOH-focused programs that have worked in other countries? Some models to draw upon?
  4. Can recommendations be outlined for different sectors to provide that multi-sectoral framework? For example, at the level of the health system, education system, community planning, etc.

Author Response

(The authors gave the same response as above.)

Round 2

Reviewer 1 Report

Thank you for the improved version of this paper.

I think the text has benefited from the further work on the data.

Nevertheless, I do think that part of my comment from the first review was not addressed. I am therefore citing it again.

"... Usually there are some models that represent these
understandings. The article stays on a too superficial level, or at least does not do the justice to
the qualitative data collected and to the topic. I feel article needs more in depth descriptions of
the processes behind the interviews and its analysis. I would prefer for the analytical part to be
more informative and transparent about, describing the ways analysis was done (from the
reference of Taylor and Bogdan it is difficult to know what was performed, for example
grounded theory, thematic analysis, IPA, ...)."

In the notes authors also report that they have used COREQ standards, but did not address that part of the review report. 

I am sure following this comment would improve the text to a further extend.

Author Response

Dear reviewer,

We thank you for your comments and for the time you took to review our manuscript. Please see the attachment, you will find more detail to your requests.

Reviewer 2 Report

This is an interesting and important paper and has been improved a lot since the initial manuscript. The methods and recruitment process, as well as support provided to participants are now clearly explained. 

There are still some issues with use of language. It is not recommended that the term 'commit suicide' be used, as the word 'commit' has connotations of crime and sin (and is considered an outdated and inappropriate description, certainly in Australia). I suggest 'attempted suicide' rather than 'attempted to commit suicide' or "die by suicide".

I'm not sure what this sentence means (line 95-6) "The relation between the research team and the participating adolescents was until the day of the interview."

I am also uncertain how the following fits within the ethical and methodological approach (lines 98-102). "The team assumed an ethical position throughout the research process, both with the interviewed persons and with mental health local authorities. In this regard, we prioritized the return of the information collected to the local authorities to be used in local public policies, through state reports,and in the publication and distribution of a book that summarizes the main results of the whole project."

Author Response

(The authors gave the same response as above.)
